# Mechanical Load-Induced Atomic-Scale Deformation Evolution and Mechanism of SiC Polytypes Using Molecular Dynamics Simulation

**DOI:** 10.3390/nano12142489

**Published:** 2022-07-20

**Authors:** Haoxiang Wang, Shang Gao, Renke Kang, Xiaoguang Guo, Honggang Li

**Affiliations:** Key Laboratory for Precision and Non-Traditional Machining Technology of Ministry of Education, Dalian University of Technology, Dalian 116024, China; wanghaoxiang@mail.dlut.edu.cn (H.W.); kangrk@dlut.edu.cn (R.K.); guoxg@dlut.edu.cn (X.G.); superhonggang@mail.dlut.edu.cn (H.L.)

**Keywords:** silicon carbide, molecular dynamics simulations, deformation mechanism, amorphous phase transformation, dislocation, stacking faults

## Abstract

Silicon carbide (SiC) is a promising semiconductor material for making high-performance power electronics with higher withstand voltage and lower loss. The development of cost-effective machining technology for fabricating SiC wafers requires a complete understanding of the deformation and removal mechanism. In this study, molecular dynamics (MD) simulations were carried out to investigate the origins of the differences in elastic–plastic deformation characteristics of the SiC polytypes, including 3C-SiC, 4H-SiC and 6H-SiC, during nanoindentation. The atomic structures, pair correlation function and dislocation distribution during nanoindentation were extracted and analyzed. The main factors that cause elastic–plastic deformation have been revealed. The simulation results show that the deformation mechanisms of SiC polytypes are all dominated by amorphous phase transformation and dislocation behaviors. Most of the amorphous atoms recovered after completed unload. Dislocation analysis shows that the dislocations of 3C-SiC are mainly perfect dislocations during loading, while the perfect dislocations in 4H-SiC and 6H-SiC are relatively few. In addition, 4H-SiC also formed two types of stacking faults.

## 1. Introduction

Silicon carbide (SiC) is a typical representative of the wide band-gap semiconductor material, which is also known as a third-generation semiconductor material. It has a high breakdown electric field, high thermal conductivity, high saturation electron velocity, and high radiation resistance [1,2,3]. It is an important material for substrate and epitaxial in wafers. In addition, SiC also has a wide application prospect in the fields of optical mirrors and biomedical devices [4,5]. At present, more than 250 kinds of SiC polytypes have been found, of which 3C-SiC, 4H-SiC, and 6H-SiC are the most widely used. The stacking sequence of the tetrahedrally bonded Si–C bilayers for 3C-SiC, 4H-SiC, and 6H-SiC are ABC/ABC…, ABCB/ABCB…, and ABCACB/ABCACB…, respectively. The different stacking sequence leads to different mechanical properties of SiC polytypes, which determine various application situations. To ensure the reliability, durability, and performance of SiC wafer or components, low surface roughness and low subsurface damage depth in nano-scale are necessary, which can be realized by ultra-precision machining, such as grinding, lapping, and polishing [6,7]. Therefore, it is of great significance to deeply study the elastic–plastic deformation mechanism of SiC single crystal in nano-scale.

Nanoindentation test is an important method to study the mechanical properties and deformation mechanism in nano-scale and has been widely used in brittle materials. Zhao et al. [8] researched the initial plastic deformation of 3C-SiC at room temperature under contact load by nanoindentation test and transmission electron microscope (TEM). Goel et al. [9] studied the nanomechanical response of 4H-SiC through a quasi-static nanoindentation test, and the critical shear stress of elastic–plastic transition is about 21 GPa. Pan et al. [10] compared the critical load of elastic–plastic transition between C-face and Si-face in 6H-SiC. The results showed that the elastic–plastic transformation threshold of the Si-face was lower. The above studies show that there are great differences in the mechanical properties of SiC polytypes. However, the internal mechanism has not been revealed due to the limitations of the experimental scale and the observation accuracy of in-situ TEM technology. The experimental observation results have difficulty ensuring the reliability of the deformation mechanism at the nano-scale. Therefore, it is of vital importance to deeply explore the atomic-scale deformation mechanism of SiC polytypes.

Molecular dynamics (MD) can simulate the trajectory and interaction of all atoms in the model, so it can deeply reveal the deformation mechanism of SiC. Sun et al. [11] used MD simulation to research the formation mechanism of prismatic dislocation rings on (111) and (110) planes of 3C-SiC during nanoindentation. Zhu et al. [12] used the MD method to simulate the deformation of the diamond indenter during 3C-SiC indentation. Zhu et al. [13] also carried out the MD simulation of nanoindentation for 4H-SiC. They found the structural phase transition to 3C-SiC and declared that the plastic deformation of 4H-SiC is affected by amorphization, dislocation glide and propagation, and stacking fault. Tian et al. [14] compared the nanoindentation characteristics of 4H-SiC and 6H-SiC on C-face and Si-face by MD simulation and experiment. The results show that the deformation mechanisms of 4H-SiC and 6H-SiC are almost the same. The hexagonal pattern in the horizontal cross-section view is composed of two symmetrical triangles centered on each other. These amorphous patterns may be caused by the gliding system of 4H-SiC and 6H-SiC. Besides, C-face is easier to form dislocations on the subsurface than on the Si-face. Wu et al. [15] used MD simulation to study the dislocation nucleation and evolution mechanism of 6H-SiC on three main planes and found that there were differences in the dislocation evolution mechanism of different crystal planes. Further, Wu et al. [16] systematically studied the amorphization and dislocation evolution mechanism of single crystal 6H-SiC in combination with the nanoindentation experiment, high-resolution transmission electron microscope (HRTEM), MD simulation, and generalized stacking failure (GSF) energy surface analysis. The results show that the plastic deformation of 6H-SiC under nanoindentation is affected by amorphization and dislocation. Amorphization corresponds to the first pop-in event, and dislocation nucleation and propagation correspond to the second pop-in event. Mishra and Szlufarsks [17] used MD simulation to determine whether 3C-SiC and under what conditions high-pressure phase transformation (HPPT) can occur during nanoindentation. The results show that HPPT induced by nanoindentation is unlikely to occur in 3C-SiC, and dislocation plasticity is the most likely mechanism of ductile deformation in 3C-SiC nanoindentation and nanomachining. Xue et al. [18] systematically studied the structural anisotropic deformation mechanism of 4H-SiC Films on different planes through nanoindentation. MD simulation results show that the formation of prismatic dislocation half-ring {1-100} on the base plane (0001) can be attributed to the interaction between a complete dislocation ring on the base plane and two complete dislocation rings on the prism plane, while prismatic dislocations circulate on the (11-20) plane and the (1-100) plane is formed by a “lasso” mechanism. Wu et al. [19] systematically discussed the effect of oxide film on 6H-SiC deformation mechanism and mechanical properties through nanoindentation MD simulation. The results show that the amorphous layer on the surface of residual indentation mark is composed of amorphous SiO_2_ and SiC. Zhao et al. [20] studied the elastic–plastic deformation mechanism of 3C-SiC in spherical nanoindentation by MD simulation. The simulation results show that before the "pop-in" event related to plastic initiation, each indented-single crystal 3C-SiC will experience a pure quasi-elastic deformation controlled by the formation of the amorphous phase. This amorphization process is completely reversible for small indentation depth. The above research has made great contributions to revealing the elastic–plastic deformation mechanism of SiC polytypes. However, the plastic deformation mechanism of SiC single crystal is still controversial, including structural phase transitions and dislocation behaviors. In addition, there is no research to systematically compare the differences of elastic–plastic deformation between SiC polytypes. These differences caused by the atomic-scale mechanism are especially still unknown and need to be deeply and systematically studied.

In this paper, the deformation properties of 3C-SiC, 4H-SiC, and 6H-SiC are compared by MD simulation. The amorphous phase transition, dislocation evolution and stacking faults of SiC polytypes are comprehensively analyzed before and after unloading. The atomic-scale deformation mechanisms of the SiC polytypes are clarified, which will provide a reference for the subsequent study of the damage evolution mechanism of SiC single crystal in nano-machining.

## 2. Materials and Methods

### 2.1. Simulation Models

All MD simulations in this paper were performed in the Large-scale Atomic/Molecular Massively Parallel Simulator (LAMMPS) [21]. Figure 1 shows the model’s initial state, and the size of the SiC workpiece was 25 nm × 25 nm × 15 nm. The radius of the spherical diamond indenter was 5 nm and 1 nm from the upper surface of SiC. The pressing direction was along the negative direction of *Z*-axis. The SiC workpiece w divided into three parts [22]: boundary, thermostatic, and Newtonian. The thickness of the boundary layer and thermostatic layer was 1 nm. The introduction of the thermostatic layer was to conduct the heat generated in the indentation process. The atomic velocity of this layer needs to be scaled to keep the temperature in this area constant. The atoms in the boundary layer remain stationary to reduce the boundary effect and eliminate the rigid motion of the model. Before indentation, the conjugate gradient (CG) method was used to minimize the energy of all models, and then the canonical ensemble (NVT) was relaxed at 300 K to simulate the nanoindentation experiment at room temperature. The micro-canonical ensemble (NVE) was used during the whole indentation process, and the atoms in the Newtonian layer abide by Newton’s second law [23]. 3C-SiC has the zinc blende structure and 4H-SiC and 6H-SiC have the wurtzite structure, as shown in Figure 1b–d. The main parameters of the models are shown in Table 1.

### 2.2. Interatomic Potential

Appropriate potentials are an important guarantee for MD simulation. Tersoff potential [24] and Vashishta potential [25] are widely used in Si and C atomic simulations and have been successfully applied in monocrystalline nanoindentation simulation [11,12,13,14,15,16,17,18,19,20] and nanoscratching simulation [26]. Tersoff potential is mostly used in nano-cutting MD simulation of SiC, while Vashishta potential is more suitable for nanoindentation MD simulation [11,13,27,28,29]. Tersoff potential also has wide applications in describing intermolecular interactions in multi-body MD simulation and is improved to ABOP potential with the further development by Erhart and Albe [30].

The total energy *E* of ABOP potential can be calculated as:(1)$$E=\sum_{i>j}f_{C}\left(r_{ij}\right)\left[V_{R}\left(r_{ij}\right)-\frac{b_{ij}+b_{ji}}{2}V_{A}\left(r_{ij}\right)\right],$$
where *f_C_*(*r*), *b_ij_*, *V_R_*(*r*), and *V_A_*(*r*) are cutoff functions, bond-order, pairwise attractive contributions, and pairwise repulsive contributions.

The Vashishta potential consists of two- and three-body interactions and is given as Equations (2)–(4):(2)$$\begin{array}{l} V=\sum_{i<j}V_{ij}^{\left(2\right)}\left(r_{ij}\right)+\sum_{i,j<k}V_{ijk}^{\left(3\right)}\left(r_{ij},r_{ik}\right)\\ =\sum_{i<j}\left(\frac{H_{ij}}{r^{\eta_{ij}}}+\frac{Z_{i}Z_{j}}{r}e^{-r/\lambda }-\frac{D_{ij}}{2r^{4}}e^{-r/\xi }-\frac{W_{ij}}{r^{6}}\right)+\sum_{i,j<k}\left(R^{\left(3\right)}\left(r_{ij},r_{ik}\right)P^{\left(3\right)}\left(\theta_{jik}\right)\right) \end{array}$$
(3)$$R^{\left(3\right)}\left(r_{ij},r_{ik}\right)=B_{jik}\exp\left(\frac{\gamma }{r_{ij}-r_{0}}+\frac{\gamma }{r_{ik}-r_{0}}\right)\Theta \left(r_{0}-r_{ij}\right)\Theta \left(r_{0}-r_{ik}\right)$$
(4)$$R^{\left(3\right)}\left(r_{ij},r_{ik}\right)=B_{jik}\exp\left(\frac{\gamma }{r_{ij}-r_{0}}+\frac{\gamma }{r_{ik}-r_{0}}\right)\Theta \left(r_{0}-r_{ij}\right)\Theta \left(r_{0}-r_{ik}\right),$$
where *H_ij_* is the strength of the steric repulsion, *Z_i_* the effective charge, *D_ij_* the strength of the charge dipole attraction, *W_ij_* is the van der Waals interaction strength, *η_ij_* the exponents of the steric repulsion term, *r_ij_* = |*r_i_* – *r_j_*| is the distance between the *i*th atom at position *r_i_* and the *j*th atom at position *r_j_*, and *λ* is the screening lengths for Coulomb, *ξ* is the charge-dipole terms, *B_jik_* is the strength of the interaction, *θ_jik_* is the angle formed by *r_ij_* and *r_ik_*, *C_jik_* and *θ_jik_* are constants, and Θ(*r*_0_ − *r_ij_*) is the step function, respectively.

In this paper, Vashishta potential was applied for the C–C, Si–Si, and C–Si atomic interaction in SiC, and ABOP potential was applied for the atomic interaction between diamond and SiC. The diamond indenter was set as a rigid body with no internal force.

### 2.3. Indentation Parameters

In order to simulate the process of ‘loading-maintaining-unloading’ in the nano indentation process, we divided the movement of the indenter into three parts. The spherical indenter was firstly indented by 100 ps to the depth of 5 nm, then maintained by 10 ps, and last was unloaded 100 ps to the initial height, as shown in Figure 2. In order to save simulation time, the moving speed of the indenter was 50 m/s, and the speed of 50 m/s has also been widely recognized in previous simulation studies [11,12,13,14,15,16,17]. The tip of the indenter used in the actual experiment (whether triangular cone or quadrangular cone) was nano spherical. Therefore, the radius of the spherical indenter selected in this paper was 5 nm. Indentation simulations were selected on the surfaces of (001), (0001) and (0001) for 3C-SiC, 4H-SiC and 6H-SiC, respectively. See Table 2 for detailed indentation parameters. In addition, the periodic boundary condition was selected in X and Y directions, and the free boundary condition was selected in the Z direction. The time step in the simulation was set to 1 fs. The Open Visualization Tool (OVITO) was used to visually analyze the results of the simulations [31]. The ‘identify diamond structure’ method in OVITO was used to identify the atomic structure [32]. Dislocations were extracted by the function of ‘dislocation analysis’ (DXA) [33]. Furthermore, coordination numbers were extracted by the function of “coordination analysis” and the cutoff radius was set to 2.5 Å. The spherical indenter was firstly indented by 100 ps at a constant speed of 50 m/s, then maintained by 10 ps when the indentation depth reached 5 nm, and last unloaded by 100 ps at a speed of 50 m/s, as shown in Figure 2. See Table 2 for detailed indentation parameters. In addition, the periodic boundary condition was selected in X and Y directions, and the free boundary condition was selected in the Z direction. The time step in the simulation was set to 1 fs.

## 3. Results and Discussion

### 3.1. Amorphous Phase Transformation

Figure 3 is a cross-sectional view of SiC polytypes at completed load and completed unload. The chosen cross-sectional planes in this paper are (100) in 3C-SiC, (1-210) in 4H-SiC, and (1-210) in 6H-SiC, respectively. It can be observed from the figure that a thick amorphous phase is formed on the surface of loaded SiC polytypes. The specific comparison shows that the amorphous phase of 3C-SiC is mainly concentrated directly below the indentation, showing a V-shape. The amorphous phases of 4H-SiC and 6H-SiC tend to expand to the sides, especially 6H-SiC. After complete unload, the residual depth of indentation of the SiC polytypes rebounded by 1.5 nm, 1.8 nm, and 1.6 nm, respectively. The distribution depth of amorphous phase is greatly reduced. Interestingly, stacking faults are only observed in 4H-SiC, which will be analyzed in depth in Section 3.3.

At the atomic level, the pair correlation function (PDF, also known as the radial distribution function) is an important method for analyzing the amorphization of crystals. PDF mainly characterizes the stacking state by the distance distribution between atoms, which is also known as *g*(*r*) function. The PDF between two particles is defined as follows:(5)$$g\left(r\right)=\frac{N\left(r\right)}{\rho 4\pi r^{2}dr},$$
where *N*(*r*) is the number of atoms within a distance of *r* and *r* + d*r* and *ρ* is the number density of atoms. Figure 4 shows the comparison of *g*(*r*) curves for perfect lattice and the lattice at completed load and completed unload. As shown in the figure, the first peaks of 3C-SiC and 4H-SiC appear at 1.88 Å, and the first peaks of 6H-SiC all appear at 1.90 Å, indicating that the Si–C bond lengths of zinc-blende and wurtzite are basically the same. Compared with the *g*(*r*) curve of perfect lattice, the *g*(*r*) curve of SiC polytypes at completed load and completed unload shows a significant decrease in peak value, which means that the number of Si–C bonds of 1.88 Å is reduced. At the same time, the characteristic peak of the *g*(*r*) curve becomes wider, indicating that the bond length of the C–Si bond changes. In other words, part of the crystal structure of silicon carbide is damaged. In addition, the peak value of *g*(*r*) curve of SiC polytypes rebounded to a certain extent after unloading, which indicates that the number of Si–C bonds of 1.88 Å is increased compared with that under loading. That means part of the crystal damage of silicon carbide has been recovered. It can be seen from Figure 3 that a large number of amorphous atoms appear under the indenter during loading, and the amorphous atoms are greatly reduced after unloading, which is consistent with the *g*(*r*) curve.

In order to further compare the degree of amorphous phase transformation in silicon carbide during loading and unloading, the number of amorphous atoms is counted in Figure 5. It counts the number of amorphous atoms in the workpiece directly below the indenter. It can be seen that the number of amorphous atoms after unloading decreases sharply compared with that during loading. Moreover, 3C-SiC has the least number of residual amorphous atoms, and 4H-SiC and 6H-SiC have more residual amorphous atoms.

The formation of amorphous phase transformation has been fully proved in previous studies [12,13,16]. The elastic recovery after unloading will affect the material removal rate in processing, which is very worthy of further research. To further explain the elastic recovery, the atomic coordination numbers (CN) at completed load and completed unload are compared, as shown in Figure 6. The CN of silicon carbide with the perfect lattice is 4. The atoms with CN < 4 are mainly distributed on the surface of the model, which is caused by the lack of 1~2 coordination atoms on the surface. During loading, a large number of atoms with CN = 5 and 6 are concentrated in the plastic deformation zone under the indenter. This is because the number of atoms in silicon carbide does not change under a high-pressure load, while the volume decreases and the atomic density increases in the plastic deformation zone, resulting in the increase of the number of atoms in the cutoff radius of the central atom. After unloading, elastic recovery and creep occur in the contact area of the indenter, and most atoms are restored to CN = 4. The change of CN often leads to amorphous phase transformation, which explains the reduction of amorphous atoms before and after unloading in Figure 5. In addition, the number of atoms with CN = 5 and 6 in 4H-SiC and 6H-SiC is significantly more than that in 3C-SiC after unloading. It is speculated that partial dislocations are more likely to cause an increase in atomic coordination numbers, which is verified in Section 3.2.

### 3.2. Dislocation Evolution

Figure 3 shows that there are a large number of dislocations on the subsurface of loaded SiC. The dislocations of 3C-SiC are mainly 1/2<110> perfect dislocations, which are scattered on both sides of the indentation, and only a small number of 1/2<110> partial dislocations exist at the edge of the amorphous layer. There are 1/3<1-210> perfect dislocations, 1/3<1-100> partial dislocations and 1/6<2-203> other dislocations in 4H-SiC and 6H-SiC at the same time. Among them, perfect dislocations are distributed on the sides of indentation, and partial dislocations and other dislocations are distributed in a disorderly manner. After complete unload, the area of dislocation distribution on the subsurface shrank to varying degrees. In order to study the dislocation evolution during indentation, the dislocation line lengths at different indentation depths during loading and unloading are counted, as shown in Figure 7. The dislocation evolution of 3C-SiC is dominated by perfect dislocation, which increases rapidly with the increase of loading, and tends to be stable after a small drop during unloading. The partial dislocation of 3C-SiC increases slowly during loading and almost remains unchanged during unloading. The perfect dislocations of 4H-SiC and 6H-SiC account for a large proportion in the loading process. However, partial dislocations and other dislocations account for much more than perfect dislocations in the unloading process.

Different lattice stacking order leads to different dislocation evolution laws. Figure 8, Figure 9 and Figure 10 show the dislocation evolution of SiC polytypes during loading. As can be seen from Figure 8a, the first dislocation appeared in 3C-SiC at *h* = 1.3 nm. 3C-SiC has only three perfect dislocations sliding along the {111} plane during *h* = 1.6–2.0 nm, and there is no overlap, as shown in Figure 8b,c. When *h* = 2.6 nm, the partial dislocations have occurred in the ellipse with the dotted line, as shown in Figure 8d. In addition, the perfect dislocation has the phenomenon of narrowing the sliding process. When the two ends of the perfect dislocation narrow and blend, it will become an independent prismatic dislocation loop separated from the damaged surface. The prismatic dislocation loop slides downward along the slip surface in the subsequent loading process, resulting in an increase in the damage depth. The dislocation evolution processes of 4H-SiC and 6H-SiC are similar. Except for perfect dislocations, a large number of partial dislocations on the (0001) plane and other dislocations on the {1-100} plane are formed on the subsurface, as shown in Figure 9 and Figure 10. Combined with the dislocation distribution while completed load and unload shown in Figure 3, the dislocations of 4H-SiC and 6H-SiC sliding downward account for a relatively small proportion, mainly sliding along the (0001) plane.

The differences in dislocation evolution mainly depend on the arrangement characteristics of crystal atoms and the direction of load. For 3C-SiC, because the {111} plane is the strong cleavage plane of 3C-SiC. The atoms along the {111} plane are densely arranged and are most prone to sliding behavior. With the increase of indentation depth, dislocation reactions on (11-1) and (111) planes were observed at *h* = 2.6 nm, accompanied by a small number of partial dislocations, circled with an elliptical dotted line in Figure 8. The dislocation transformation is carried out by dislocation decomposition and dislocation synthesis. The Burgers vector expression is as follows:1/2[1 −1 0] = 1/6[1 −2 −1] + 1/6[2 −1 1](6)
1/6[1 −2 −1] + 1/6[2 −1 1] + 1/2[0 1 −1] = 1/2[1 0 −1].(7)

The above two dislocation transformation reactions are the process of energy reduction, indicating the possibility of dislocation transformation. The former occurs on the (11-1) plane, and the decomposed partial dislocations cross slip to become partial dislocations on the (111) plane. Then, the perfect dislocation 1/2[01-1] sliding on the (111) plane blends with the partial dislocation to form perfect dislocation 1/2[10-1]. In addition, the formation and downward slip of prismatic dislocation rings have also been reported in the study of sun et al. [11]. For 4H-SiC and 6H-SiC, because the pressing direction of the indenter is perpendicular to the (0001) cleavage plane, weakening the atomic sliding ability along the (0001) plane. Moreover, the {1-100} plane is the second cleavage plane of hexagonal silicon carbide. When the load is large, it can also produce atomic slip and form dislocations.

### 3.3. Stacking Faults

Figure 3b shows that a large number of 3C cubic crystals and 2H hexagonal crystals remained on the 4H-SiC subsurface after unloading. The change of atomic stacking indicates that stacking faults occur in 4H-SiC. On the (1-210) plane, stacking faults were located in the right region and the lower-left region of the indentation, respectively, accompanied by 1/3<1-100> partial dislocations. By analyzing the sliding motion under the action of dislocation, the generation mechanism of the above two kinds of stacking faults can be explained, as shown in Figure 11. Figure 11a shows two rows of atoms of 4H-SiC on the (1-210) plane, and its atomic stacking structure is ABCB. The atoms in the lower-left region of the indentation produce 1/3[-1010] dislocation under the action of the transverse extrusion component, as shown in Figure 11b. This dislocation makes the atoms stacked at position C slip 1/3 of the lattice length in the directions of [-1000] and [10], about 1 Å. The atomic position after sliding is directly below A, and the atomic stacking structure becomes ABAB, which is a typical 2H hexagonal crystal. Similarly in Figure 11c, the atoms in the right region of the indentation are affected by 1/3[10-10] dislocation, causing the atoms stacked at position B to slide 1/3 of the lattice length in the [1000] and [0,1,2,3,4,5,6,7,8,9,10] directions. The atomic position after sliding is directly below A, and the atomic stacking structure becomes ABC, which is a typical 3C cubic crystal.

### 3.4. Temperature

Figure 12 shows the temperature change curve of SiC polytypes during the indentation process. The temperature value in the figure is the average temperature of the whole workpiece. It can be seen that the temperature change trend of the three is the same, which increases with the increase of the pressing depth, and the temperature increases faster and faster. Before *h* = 4 nm, the temperature of 3C-SiC is slightly higher than that of 4H-SiC and 6H-SiC. However, after *h* = 4 nm, it is found that the temperature rise rate of 4H-SiC after is significantly faster than that of 3C-SiC and 6H-SiC. We speculate that this is because the generation of stacking faults leads to the release of energy.

## 4. Conclusions

This work systematically compares the elastic–plastic deformation characteristics of SiC polytypes by MD simulation, and mainly analyzes the characteristics and differences of amorphous phase transformation, dislocation evolution and stacking faults. The main conclusions are as follows:(1)The change of atomic coordination number caused by the high-pressure load is the main reason for promoting amorphous phase transformation. The amorphous phase transformation runs through the whole process and recovers significantly after unloading;(2)The atomic slip in 3C-SiC is dominated by perfect dislocations and tends to develop into prismatic dislocation rings, while 4H-SiC and 6H-SiC are dominated by various dislocations;(3)Two types of stacking faults occur in 4H-SiC, which are caused by partial dislocations.

## Figures and Tables

**Figure 1 nanomaterials-12-02489-f001:**
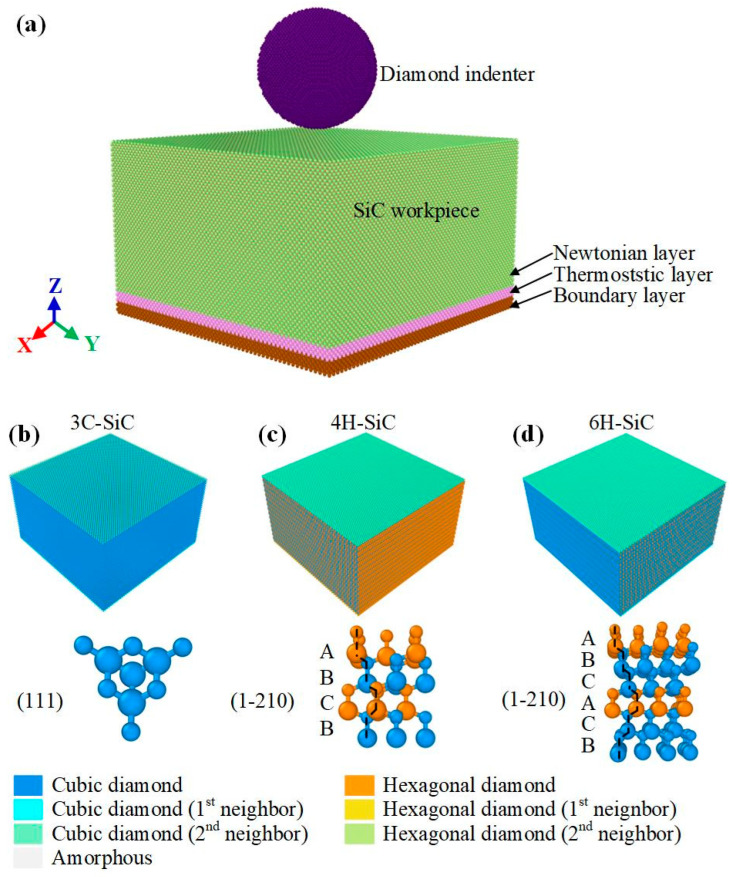
(**a**) MD simulation model of nanoindentation for three types of SiC single crystal: (**b**) 3C-SiC, (**c**) 4H-SiC, and (**d**) 6H-SiC, respectively.

**Figure 2 nanomaterials-12-02489-f002:**
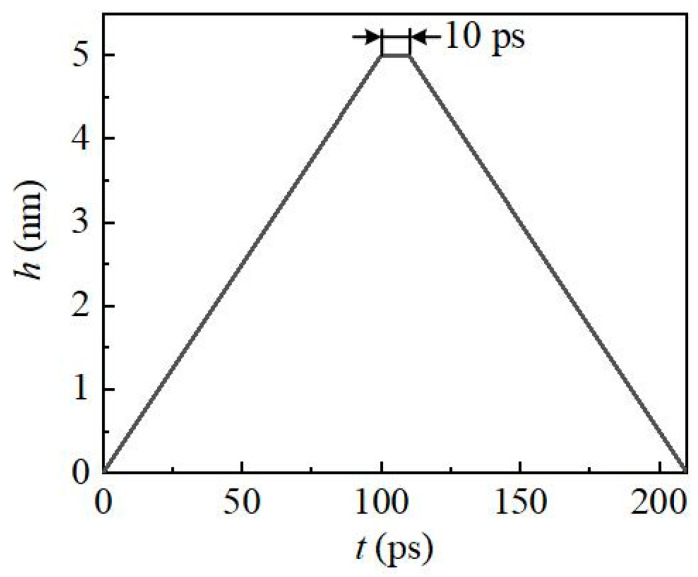
Motion parameter of diamond indenter.

**Figure 3 nanomaterials-12-02489-f003:**
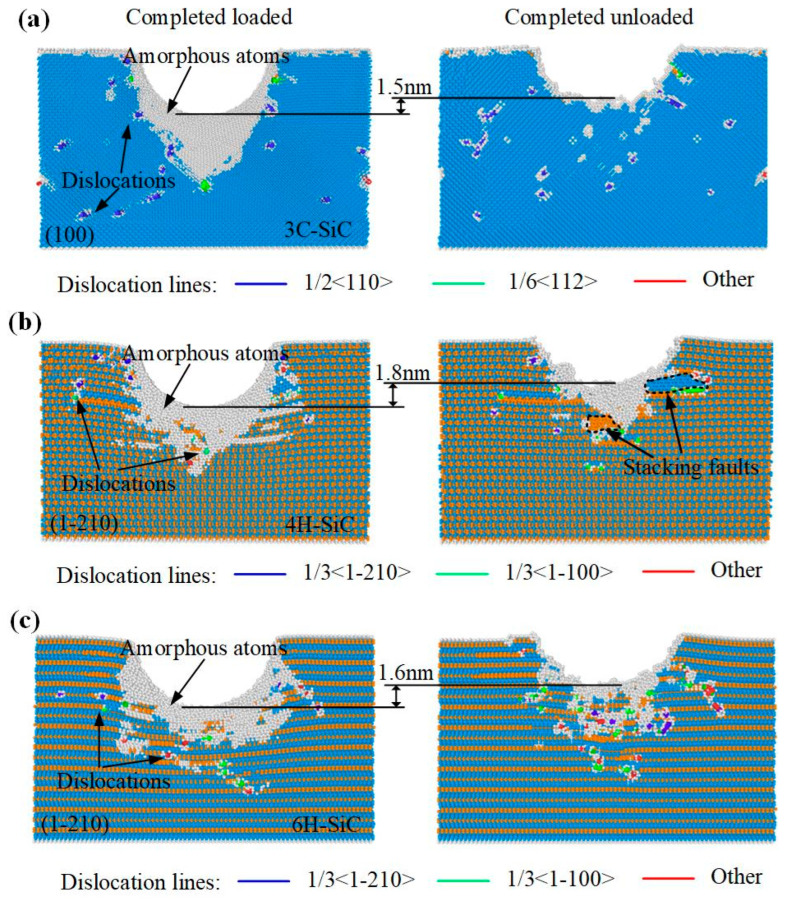
Cross-sectional view of nanoindentation for three types of SiC single crystal at completed load and completed unload. (**a**) (100) plane in 3C-SiC, (**b**) (1-210) plane in 4H-SiC, and (**c**) (1-210) plane in 6H-SiC, respectively.

**Figure 4 nanomaterials-12-02489-f004:**
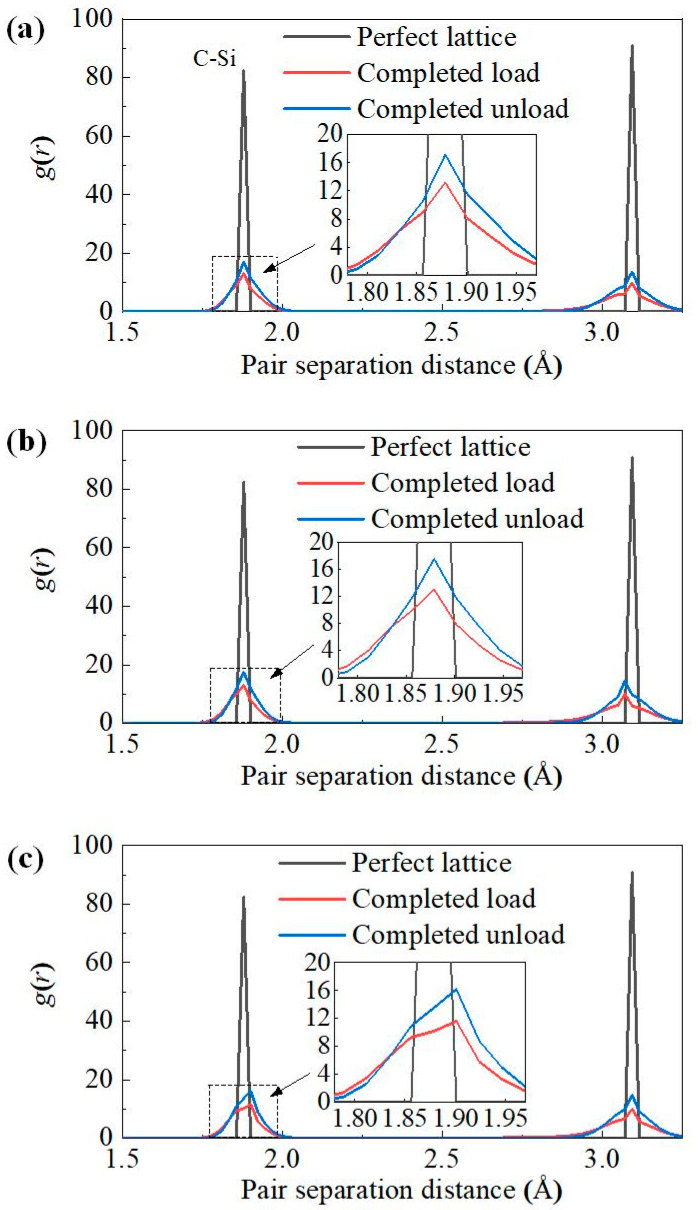
Pair correlation analysis for three types of SiC single crystal. (**a**) 3C-SiC, (**b**) 4H-SiC, and (**c**) 6H-SiC, respectively.

**Figure 5 nanomaterials-12-02489-f005:**
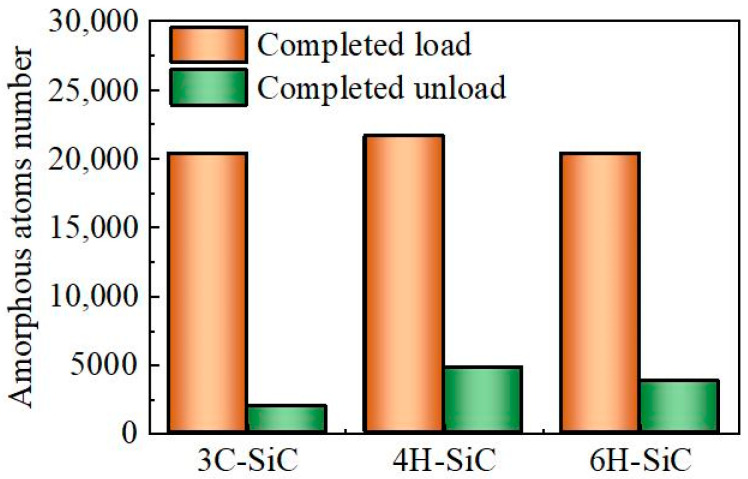
The number of amorphous atoms for three types of SiC single crystal at completed load and completed unload.

**Figure 6 nanomaterials-12-02489-f006:**
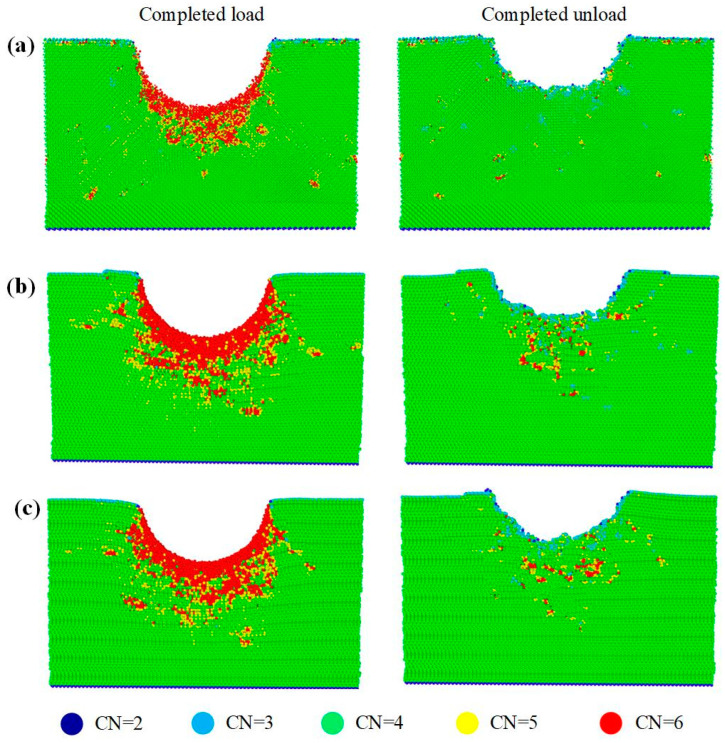
Coordination analysis for three types of SiC single crystal at completed load and completed unload. (**a**) 3C-SiC, (**b**) 4H-SiC, and (**c**) 6H-SiC, respectively.

**Figure 7 nanomaterials-12-02489-f007:**
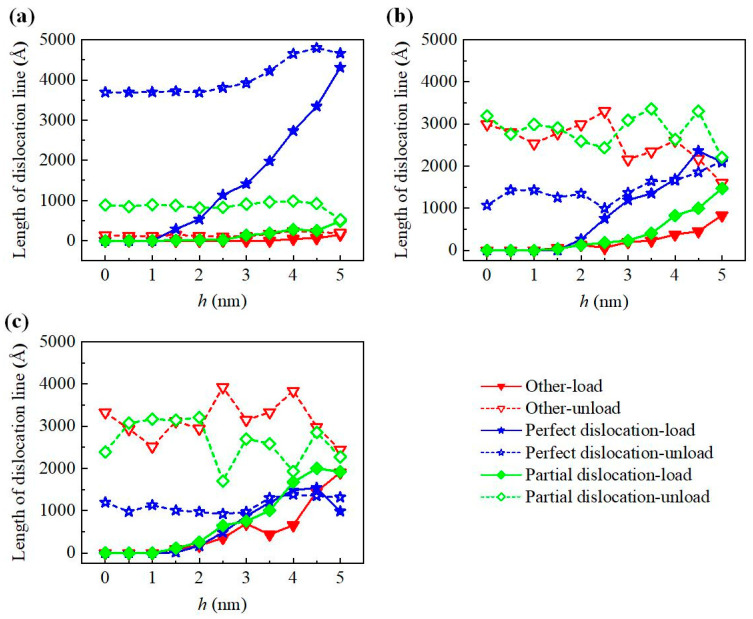
Dislocation line length of three types of SiC single crystal during loading and unloading process: (**a**) 3C-SiC, (**b**) 4H-SiC, and (**c**) 6H-SiC, respectively.

**Figure 8 nanomaterials-12-02489-f008:**
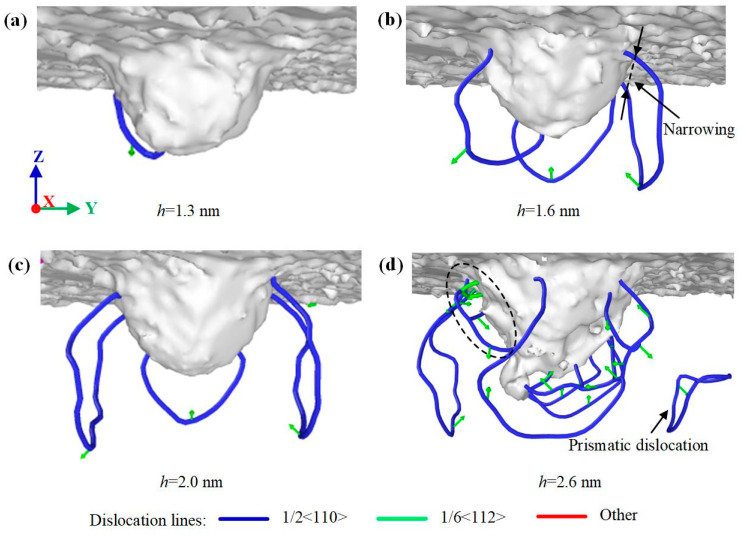
Dislocation distribution of 3C-SiC at different indentation depths: (**a**) *h* = 1.3 nm, (**b**) *h* = 1.6 nm, (**c**) *h* = 2.0 nm, and (**d**) *h* = 2.6 nm, respectively.

**Figure 9 nanomaterials-12-02489-f009:**
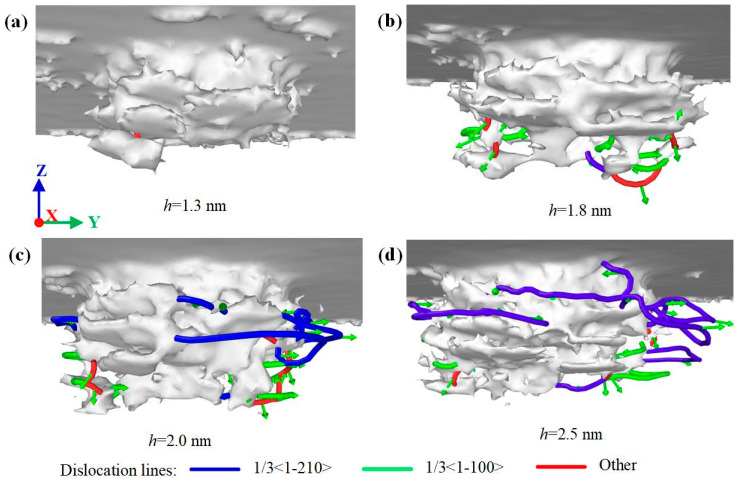
Dislocation distribution of 4H-SiC at different indentation depths: (**a**) *h* = 1.3 nm, (**b**) *h* = 1.8 nm, (**c**) *h* = 2.0 nm, and (**d**) *h* = 2.5 nm, respectively.

**Figure 10 nanomaterials-12-02489-f010:**
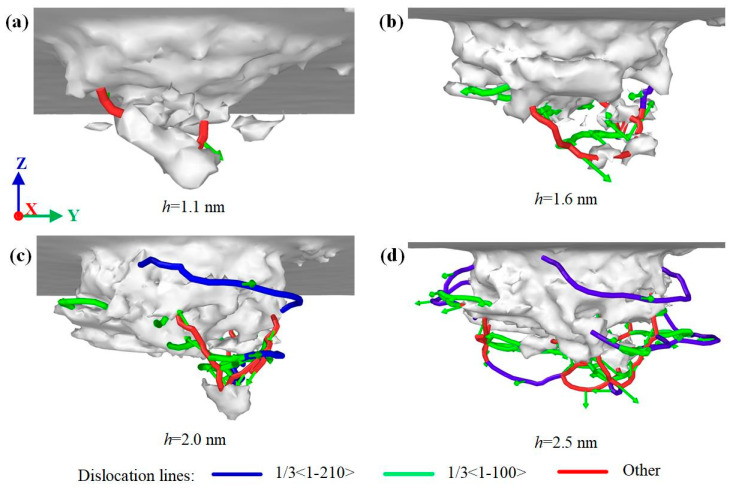
Dislocation distribution of 6H-SiC at different indentation depths: (**a**) *h* = 1.1 nm, (**b**) *h* = 1.6 nm, (**c**) *h* = 2.0 nm, and (**d**) *h* = 2.5 nm, respectively.

**Figure 11 nanomaterials-12-02489-f011:**
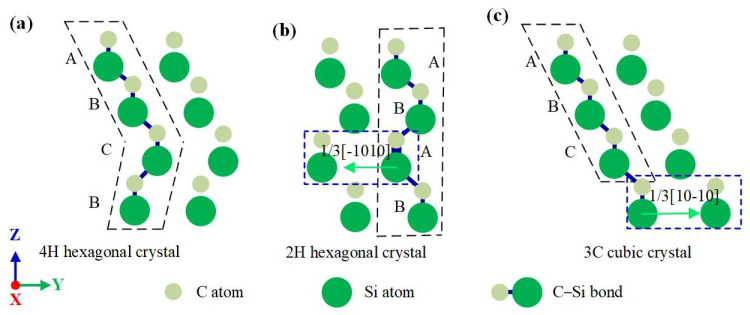
Schematic diagram of the stacking faults formation in 4H-SiC. (**a**) 4h hexagonal crystal structure, (**b**) 2H hexagonal crystal, and (**c**) 3C cubic crystal.

**Figure 12 nanomaterials-12-02489-f012:**
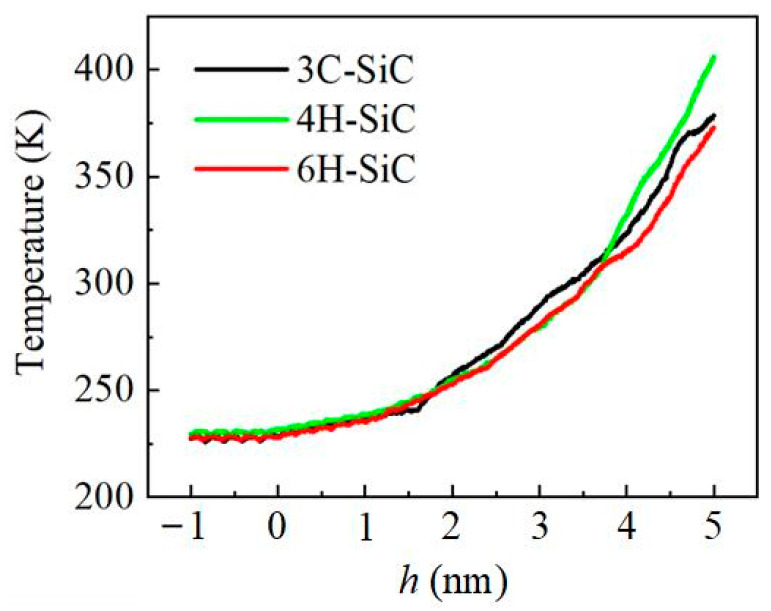
Temperature of SiC polytypes during indentation.

**Table 1 nanomaterials-12-02489-t001:** Main parameters of the simulation models.

Parameters	Value
Dimensions of specimens	25 nm × 25 nm × 15 nm
Radius of indenter	5 nm
Number of atoms	About 1,000,000
Lattice constants (Å)	3C-SiC: *a* = 4.360. 4H-SiC: *a* = 3.073, *c* = 10.053. 6H-SiC: *a* = 3.095, *c* = 15.170
Relaxation ensemble	NVT
Ensemble	NVE

**Table 2 nanomaterials-12-02489-t002:** Main parameters of indenting.

Parameters	Value
Indentation depth	5 nm
Indentation speed	50 m/s
Unload speed	50 m/s
Indentation surface	3C-SiC: (0 0 1), 4H-SiC: (0 0 0 1), 6H-SiC: (0 0 0 1)
Indenting direction	3C-SiC: [0 0 −1], 4H-SiC: [0 0 0 −1], 6H-SiC: [0 0 0 −1]
Equilibration temperature	300 K
Timestep	1 fs

## Data Availability

Not applicable.

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
