# Peer review of "Mechanical Load-Induced Atomic-Scale Deformation Evolution and Mechanism of SiC Polytypes Using Molecular Dynamics Simulation"

_nanomaterials, 2022, doi:10.3390/nano12142489_

Round 1
Reviewer 1 Report
Authors reported a comprehensive and original investigation on the origins of the differences in elastic-plastic deformation characteristics of the SiC polytypes during nanoindentation. 3C-SiC, 4H-SiC and 6H-SiC crystals have been considered. They performed LAMMPS molecular dynamics simulations with 1000000 atoms for the SiC workpiece and diamond nanoindenter interaction. The analysis of simulation results shows that the deformation mechanisms of SiC polytypes are all dominated by amorphous phase transformation and dislocation behaviors.
The manuscript reports significant novelty and advance in the knowledge of the mechanical properties and deformation mechanism of SiC during nanoindentation.
The paper is original and the current literature of material science can benefit from the submitted work. Figures and references properly integrate the whole manuscript. All sections are well argued. I suggest to publish the paper in the MDPI Nanomaterials journal.
Below you can find some feedbacks which could help the manuscript reading:
1. Pag. 3, Sec. 2: Authors should describe how the data reported in Sec. 3 have been extracted from the various MD snapshots. In particular how amorphous atoms and dislocations are detected in Fig. 3, how amorphous atoms are detected in Fig. 5, how the coordination numbers are extracted in Fig. 6, how dislocation line lengths are calculated in Fig. 7. This can support the reproducibility of the data analysis.
2. Pag. 5, Section 2.3 Indentation parameters: Authors should explain the particular choice of the indentation parameters, that are the indentation speed, indentation depth, probe radius and the exposed surfaces for the various polytypes.
3. Pag. 5, Section 2.3 Indentation parameters: A reader could be interested to see the deformation mechanisms for various indentation speeds, for example one order of magnitude less and more, that are 5 m/s and 500 m/s with the same indentation depth of 5 nm. This referee strongly suggests to add this analysis in the manuscript, following a similar study reported in Sec. 3 and making proper comparisons with varying indentation speeds.
4. Pag. 6, Fig. 3: Authors should report in the figure caption and the main text where Fig. 3 is discussed the cross sectional plane chosen for Fig. 3a,b,c.
5. Pag. 6, line 185: Sentence "N(r) is the number of atoms within the cutoff radius r". This sentence is not correct. N(r) is the number of particles within a distance of r and r+dr away from a particle (https://en.wikipedia.org/wiki/Radial_distribution_function). Authors should check the formal definition of the radial distribution function.
6. Pag. 10, Fig. 8,9,10: Authors should show more snapshots during the indentation process. This could help to follow in a continuous way the evolution of various dislocations during the loading and unloading of the nanoindenter.
Reviewer 2 Report
My comments are provided in the attached pdf.

Round 2
Reviewer 1 Report
Authors addressed all points raised by the reviewer. I suggest to publish the paper in the current form in the MDPI Nanomaterials journal.